# Elastic-InfoGAN: Unsupervised Disentangled Representation Learning in Class-Imbalanced Data

Utkarsh Ojha[1]    Krishna Kumar Singh[1,2]    Cho-Jui Hsieh[3]    Yong Jae Lee[1]

[1]UC Davis        [2]Adobe Research        [3]UCLA

utkarshojha.github.io/elastic-infogan/

## Abstract

We propose a novel unsupervised generative model that learns to disentangle object identity from other low-level aspects in class-imbalanced data. We first investigate the issues surrounding the assumptions about uniformity made by InfoGAN [10], and demonstrate its ineffectiveness to properly disentangle object identity in imbalanced data. Our key idea is to make the discovery of the discrete latent factor of variation invariant to identity-preserving transformations in real images, and use that as a signal to learn the appropriate latent distribution representing object identity. Experiments on both artificial (MNIST, 3D cars, 3D chairs, ShapeNet) and real-world (YouTube-Faces) imbalanced datasets demonstrate the effectiveness of our method in disentangling object identity as a latent factor of variation.

## 1   Introduction

Generative models aim to model the true data distribution, so that *fake* samples that seemingly belong to the modeled distribution can be generated [1, 42, 6]. Recent deep neural network based models such as Generative Adversarial Networks [19, 44, 43] and Variational Autoencoders [33, 24] have led to promising results in generating realistic samples for high-dimensional and complex data such as images. More advanced models show how to discover *disentangled* (factorized) representations [57, 10, 49, 26, 47], in which different latent dimensions can be made to represent independent factors of variation (e.g., pose, identity) in the data (e.g., human faces).

InfoGAN [10] in particular, learns an unsupervised disentangled representation by maximizing the mutual information between the discrete or continuous latent variables and the corresponding generated samples. For discrete latent factors (e.g., digit identities), it assumes that they are uniformly distributed in the data, and approximates them accordingly using a *fixed uniform* categorical distribution. Although this assumption holds true for many benchmark datasets (e.g., MNIST [34]), real-word data often follows a long-tailed distribution and rarely exhibits perfect balance between the categories. Indeed, applying InfoGAN on imbalanced data can result in incoherent groupings, since it is forced to discover potentially non-existent factors that are uniformly distributed in the data; see Fig. 1.

In this work, we augment InfoGAN to discover disentangled categorical representations from *imbalanced* data. Our model, Elastic-InfoGAN, makes two improvements to InfoGAN which are simple and intuitive. First, we remodel the way the latent distribution is used to fetch the latent variables; we lift the assumption of any knowledge about the underlying class distribution, where instead of deciding and fixing them beforehand, we treat the class probabilities as *learnable parameters* of the optimization process. To enable the flow of gradients back to the class probabilities, we employ the Gumbel-Softmax distribution [30, 36], which acts as a proxy for the categorical distribution,

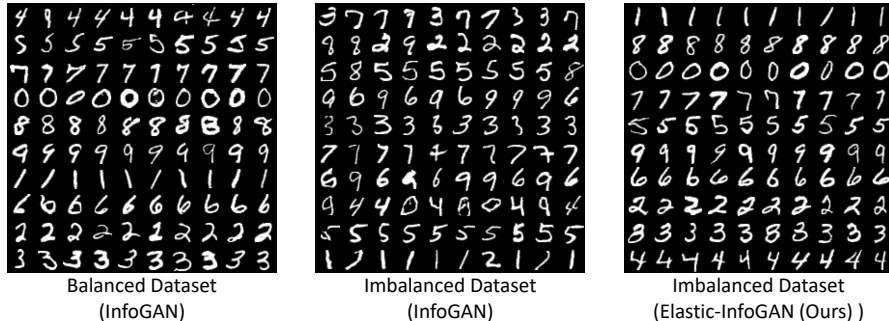

| Balanced Dataset | Imbalanced Dataset | Imbalanced Dataset |
| (InfoGAN) | (InfoGAN) | (Elastic-InfoGAN (Ours) ) |

Figure 1: (**Left** & **Center**) Samples generated with an InfoGAN model learned with a fixed uniform categorical distribution $Cat(K = 10, p = 0.1)$ on balanced and imbalanced data, respectively. Each row corresponds to a different learned latent category. (**Right**) Samples generated with Elastic-InfoGAN using its automatically learned latent categorical distribution. Although InfoGAN discovers digit identities in the balanced data, it produces redundant/incoherent groupings in the imbalanced data. In contrast, our model is able to discover digit identities in the imbalanced data.

generating *differentiable* samples having properties similar to that of categorical samples. Our second improvement stems from an observation of a failure case of InfoGAN (Fig. 1 center); we see that the model has trouble generating consistent images from the same category for a latent dimension (e.g., rows 1, 2, 4). This indicates that there are other low-level factors (e.g., rotation, thickness) which the model focuses on while categorizing the images. Although there are multiple meaningful ways to partition unlabeled data—e.g., with digits, one partitioning could be based on identity, whereas another could be based on stroke width—we aim to discover the partitioning that groups objects according to a high-level factor like identity while being invariant to low-level "nuisance" factors like lighting, pose, and scale changes. To this end, we take inspiration from self-supervised contrastive representation learning literature [4, 23, 9] to learn representations focusing on *object identity*. Specifically, we enforce (i) similar representations for positive pairs (e.g., an image and its mirror-flipped version), and (ii) dissimilar representations for negative pairs (e.g., two different images). As a result, the discovered latent factors align more closely with object identity, and less with other factors. Such partitionings focusing on object identity are more likely to be useful for downstream visual recognition applications; e.g. (i) semi-supervised object recognition [43, 41] or image retrieval using object-identity based image features; (ii) performing data augmentation to remove class-imbalance using synthetic images.

Importantly, Elastic-InfoGAN retains InfoGAN's ability to jointly model both continuous and discrete factors in either balanced or imbalanced data scenarios. To our knowledge, our work is the first to tackle the problem of disentangled representation learning in the scenario of imbalanced data, *without* the knowledge of ground-truth class distribution (Fig. 1 right). We show qualitatively and quantitatively our superiority in terms of the ability to disentangle object identity as a factor of variation, in comparison to relevant baselines. And in order to discover object identity as a factor, our results also provide interesting observations regarding the *ideal* distribution for the latent variables.

## 2   Related Work

**Disentangled representation learning**  has a vast literature [25, 5, 57, 10, 38, 49, 12, 26, 47]. In particular, InfoGAN [10] learns disentanglement without supervision by maximizing the mutual information between the latent codes and generated images, and has shown promising results for *class-balanced* datasets like MNIST [34], CelebA [35], and SVHN [39]. JointVAE [15] extends beta-VAE [24] by jointly modeling both continuous and discrete factors, using Gumbel-Softmax sampling. However, both InfoGAN and JointVAE assume uniformly distributed data, and hence fail to be equally effective in imbalanced data, evident by Fig. 1 and our experiments. Our work proposes improvements to InfoGAN to enable it to discover meaningful latent factors in *imbalanced* data.

**Learning from imbalanced data**  Real world data have a long-tailed distribution [20, 50], which can impede learning, since the model can get biased towards the dominant categories. Re-sampling [8, 22, 46, 7, 60] and class re-weighting techniques [48, 28, 13, 37] can alleviate this issue for the

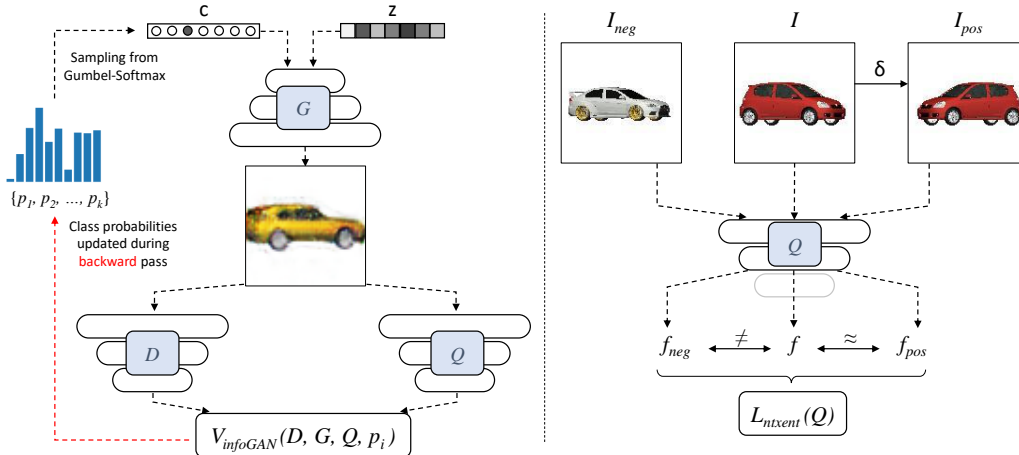

Figure 2: (**Left**) Our model takes a sampled categorical code from a Gumbel-Softmax distribution and a noise vector to generate fake samples. The use of differentiable latent variables from the Gumbel-Softmax enables gradients to flow back to the class probabilities to update them. (**Right**) Apart from the original InfoGAN [10] losses, we have an additional loss for contrastive learning of representations. We transform a real image $I$ using commonly used data augmentations $\delta$ (e.g. mirror flipping, random crop) to create a positive pair $\{I, I_{pos}\}$, and enforce similarity in their respective features extracted using $Q$. The same real image is also paired up with an arbitrary image from the same batch to create a negative pair $\{I, I_{neg}\}$, whose feature representations are made dissimilar.

*supervised* setting, in which the class distributions are known a priori. There are also unsupervised clustering methods that deal with imbalanced data in unknown class distributions (e.g., [40, 59]). Our model works in the same *unsupervised* setting; however, unlike these methods, we propose a *generative* model method that learns to disentangle latent categorical factors in imbalanced data.

**Data augmentation for unsupervised image grouping** Unsupervised deep clustering methods [18, 58, 45, 54] try to group unlabeled instances that belong to the same object category. Some works [29, 14, 27, 31, 4, 9] use data augmentation for image transformation invariant clustering or representation learning. The main idea is to maximize the mutual information or similarity between the features of an image and its corresponding transformed image. Some approaches also try to make different images more dissimilar [4, 9]. However, unlike our approach, these methods typically do not target imbalanced data and do not perform generative modeling.

**Differentiable approximation of categorical variables** Existence of discrete random variables within a computation graph introduces non-differentiability. Consequently, recent works have introduced a reparameterization trick using Gumbel-Softmax, enabling differentiable sampling of variables which approximate the categorical ones [30, 36]. This has been used in neural architecture search [52, 55, 53], where it makes the process of choosing blocks (e.g. out of $k$ distinct options) for a layer differentiable, enabling the overall search process possible through gradient-based optimization methods. Recently, it has also found use-cases in approximating discrete segmentation masks for scene generation [3]. Our work uses Gumbel-Softmax reparameterization for a different application, where we seek to learn a better multinomial distribution for discrete factors (e.g. object identity) in class-imbalanced data.

## 3 Approach

Let $\mathcal{X} = \{x_1, x_2, \ldots, x_N\}$ be a dataset of $N$ unlabeled images from $k$ different classes. No knowledge about the nature of class imbalance is known beforehand. Our goal is to learn a generative model $G$ which can learn to disentangle object category from other aspects (e.g., digits in MNIST [34], face identity in YouTube-Faces [51]) in imbalanced data, by approximating the appropriate latent distribution. In the following, we first briefly discuss InfoGAN [10], which addressed this problem for the balanced setting. We then explain how it can be improved to handle imbalanced data.

## 3.1 Background: InfoGAN

Learning disentangled representations using the GAN [19] framework was introduced in InfoGAN [10]. The intuition is for the generated samples to retain information about latent variables, and consequently for latent variables to gain control over certain aspects of the generated image. In this way, different types of latent variables (e.g., discrete categorical vs. continuous) can control properties like discrete (e.g., digit identity) or continuous (e.g., digit rotation) variations in the generated images.

Formally, InfoGAN does this by maximizing the mutual information between the latent code $c$ and the generated samples $G(z, c)$, where $z \sim P_{noise}(z)$ and $G$ is the generator network. The mutual information $I(c, G(c, z))$ can then be used as a regularizer in the standard GAN training objective. Computing $I(c, G(c, z))$ however, requires $P(c|x)$, which is intractable and hard to compute. The authors circumvent this by using a lower bound of $I(c, G(c, z))$, which can approximate $P(c|x)$ via a neural network based auxiliary distribution $Q(c|x)$. The training objective hence becomes:

$$\min_{G,Q} \max_D V_{InfoGAN}(D, G, Q) = V_{GAN}(D, G) - \lambda_1 L_1(G, Q), \qquad (1)$$

where $L_1(G, Q) = E_{c \sim P(c), x \sim G(z,c)}[\log Q(c|x)] + H(c)$, $D$ is the discriminator network, and $H(c)$ is the entropy of the latent code distribution. Training with this objective results in latent codes $c$ having control over the different factors of variation in the generated images $G(z, c)$. To model discrete variations in the data, InfoGAN employs non-differentiable samples from a uniform categorical distribution with fixed class probabilities; i.e., $c \sim Cat(K = k, p = 1/k)$ where $k$ is the number of discrete categories to be discovered.

## 3.2 Disentangling object identity in imbalanced data

As shown in Fig. 1, applying InfoGAN to an imbalanced dataset results in suboptimal disentanglement, since the uniform prior assumption does not match the actual ground-truth data distribution of the discrete factor (e.g., digit identity). To address this, we propose two improvements to InfoGAN. The first is to enable *learning* of the latent distribution's parameters (class probabilities), which requires gradients to be backpropagated through latent code samples $c$, and the second involves constrastive learning of representations, so that the discovered factor aligns closely with *object identity*.

**Learning the prior distribution** To learn the prior distribution, we replace the fixed categorical distribution in InfoGAN with the Gumbel-Softmax distribution [30, 36], which enables sampling of differentiable samples. The continuous Gumbel-Softmax distribution can be smoothly annealed into a categorical distribution. Specifically, if $p_1, p_2..., p_k$ are the class probabilities, then sampling of a $k$-dimensional vector $c$ can be done in a differentiable way:

$$c_i = \frac{\exp((\log(p_i) + g_i)/\tau)}{\sum_{j=1}^k \exp((\log(p_j) + g_j)/\tau)} \qquad \text{for } i = 1, ..., k. \qquad (2)$$

Here $g_i, g_j$ are samples drawn from $Gumbel(0, 1)$, and $\tau$ (softmax temperature) controls the degree to which samples from Gumbel-Softmax resemble the categorical distribution. Low values of $\tau$ make the samples possess properties close to that of a one-hot sample.

In theory, InfoGAN's behavior in the class balanced setting (Fig. 1 left) can be replicated in the imbalanced case (where grouping becomes incoherent, Fig. 1 center), by simply replacing the fixed uniform categorical distribution with Gumbel-Softmax with *learnable* class probabilities $p_i$'s; i.e. gradients can flow back to update the class probabilities (which are uniformly initialized) to match the true class imbalance. And once the true imbalance gets reflected in the class probabilities, the *possibility* of proper categorical disentanglement (Fig. 1 right) becomes feasible.

Empirically, however, this ideal behavior is not observed in a consistent manner. As shown in the figure on the right, unsupervised grouping can focus on non-categorical attributes such as rotation of the digit (left). Although this is one valid way to group unlabeled data, our goal is to have groupings that correspond to *class identity* 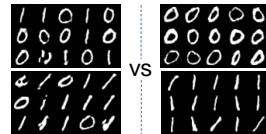 (right). This would enable useful applications for downstream tasks such as semi-supervised image classification, removing class-imbalance with generated data, and identity-based image retrieval.

**Learning object identities** Based on Eq. 1, the factor of variation discovered by the latent vector $c$ will depend on the factor that $Q$ focuses on while making the classification decision (whether images

are assigned different classes based on e.g., pose, illumination, identity, etc.). So, to enable the discovery of object identity as the factor, we enforce $Q$ to learn representations using a contrastive loss [21]. The idea is to create positive pairs (e.g., a car and it's mirror flipped version) and negative pairs (e.g., a red hatchback and a white sedan) based on object identity, and have $Q$ produce similar and dissimilar representations for them respectively.

Since we do not have any category labels, we treat each instance (image) as its own class. Intuitively, and as validated by prior work [4, 9], forcing a model to predict the same feature for two different views (augmentations) of the same image leads to the model learning a representation focusing on high-level factors like object identity. Formally, for a sampled batch of $N$ real images, we construct their augmented version, by applying *identity preserving* transformations ($\delta$) to each image, resulting in a total of $2N$ images. For each image $I_i$ in the batch, we define the corresponding transformed image as $I_{pos}$, and all other $2(N-1)$ images as $I_{neg}$. We define the following loss for each image $I_i$ in the batch, where $\{I_i, I_{pos}\}$ and $\{I_i, I_{neg}\}$ act as positive and negative pairs, respectively:

$$\ell_i = -\log \frac{\exp\left(\mathrm{sim}(f_i, f_j)/\tau'\right)}{\sum_{k=1}^{2N} \mathbf{1}_{[k \neq i]} \exp\left(\mathrm{sim}(f_i, f_k)/\tau'\right)} \tag{3}$$

where $j$ indexes the positive pair, $f$ represents the feature extracted using $Q$ (we use the penultimate layer), $\tau'$ is a softmax temperature, and sim(.) refers to cosine similarity. Note that since any two (unlabeled) images in a batch, except $I_i$ and $I_{pos}$, are treated as negative pairs, there would be cases where the sampled pair will be a false negative (i.e., images belonging to the same category). However, the fraction of false negatives remains considerably low except for highly-skewed data scenarios (we provide analysis in the supplementary), and even then we take the penultimate layer's features for computing similarity/dissimilarity which provides some robustness to such errors as the final layer can still put two false negatives to be similar to each other. These aspects make the above approximation of sampling negative pairs practically applicable in our setting. We denote the overall loss as $L_{ntxent} = \sum_{i=1}^{N} \ell_i$ (normalized temperature-scaled cross entropy loss [9]). Our training objective hence becomes:

$$\min_{G,Q,p_i} \max_D L_{final} = V_{InfoGAN}(D, G, Q, p_i) + \lambda_2 L_{ntxent}(Q). \tag{4}$$

$V_{InfoGAN}$ plays the role of generating realistic images and associating the latent variables to correspond to *some* factor of variation in the data, while the addition of $L_{ntxent}$ will push the discovered factor of variation to be close to object identity. The latent codes sampled from Gumbel-softmax, generated fake images, and losses operating on fake images are all functions of class probabilities $p_i$'s too. Thus, during the minimization phase of $L_{final}$, the gradients are used to optimize the class probabilities along with $G$ and $Q$ in the backward pass. Overall, we leverage previous ideas used in orthogonal areas: Gumbel-Softmax was introduced for differentiable sampling of one-hot like variables, and data augmentations have been used for various visual recognition tasks. Our framework integrates them in a coherent way to address the new problem of generative modeling of latent object identity factor in class-imbalanced data.

## 4 Experiments

In this section, we perform quantitative and qualitative analyses to demonstrate the advantage of our model in discovering categorical disentanglement for imbalanced datasets.

**Datasets** **(1) MNIST** [34] is by default a balanced dataset with 70k images, with a similar number of training samples for each of 10 classes. We artificially introduce imbalance over 50 random splits, and report the results averaged over them. **(2) 3D Cars** [17] and **(3) 3D Chairs** [2] consist of synthetic objects rendered with varying identity and poses. We choose 10 random object categories for both 3D cars (out of 183 total categories) and 3D chairs (out of 1396 total categories), where each category contains 96 and 62 images for cars and chairs, respectively. Within the chosen 10 categories, we introduce imbalance over 5 random splits. The whole process is repeated 5 times (choosing a different set of 10 categories randomly each time) so as to test the generalizability of the approach. **(4) ShapeNet** is a dataset of 3D models of diverse object categories, whose 2D renderings can be obtained in different pose/viewpoints. We choose 5 categories (synsets) from ShapeNetCore - cars, airplanes, bowl, can, rifle - which are more diverse in object shape/appearance compared to

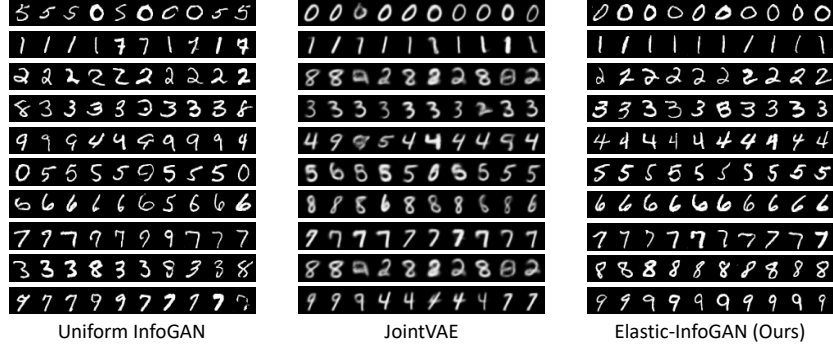

Uniform InfoGAN        JointVAE        Elastic-InfoGAN (Ours)

Figure 3: Representative generations on a random imbalanced MNIST split. Each row corresponds to a learned latent variable. Elastic-InfoGAN generates inconsistent images in only the 4th row (8 with 3's), whereas Uniform InfoGAN and JointVAE do so in many rows (e.g. rows 1, 9 and rows 3, 8 respectively).

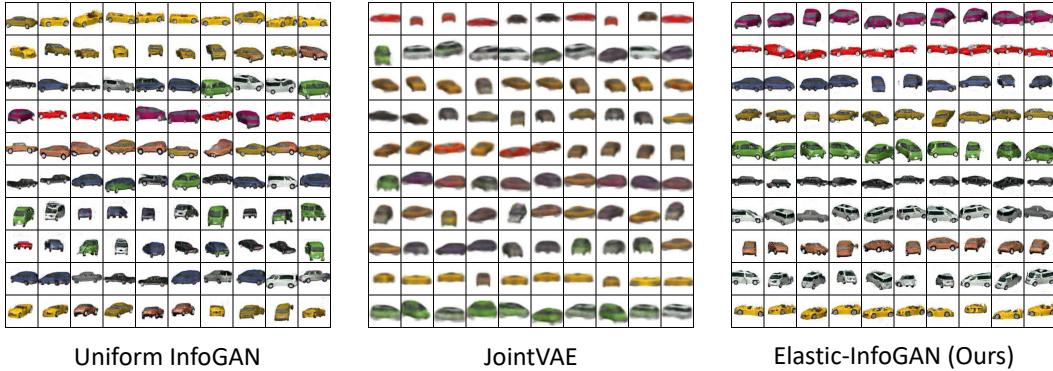

Uniform InfoGAN        JointVAE        Elastic-InfoGAN (Ours)

Figure 4: Image generations on a random imbalanced 3D Cars split, on a randomly chosen sets of categories. Each row corresponds to a learned latent variable. The images are much more consistent corresponding to a latent variable for Elastic-InfoGAN, compared with Uniform InfoGAN. JointVAE struggles in the aspects of realism as well as consistency among the generations for a latent code.

3D Cars/Chairs. Each category has a large number of instances (different car models within the *cars* synset). We select different number of instances for each of the categories to introduce imbalance, and generate 30 renderings [1] for each instance in different viewpoints. **(5) YouTube-Faces** [51] is a real world imbalanced dataset with varying number of training samples (frames) for 40 face identity classes (as used in [45]). The smallest/largest class has 53/695 images, with a total of 10,066 tightly-cropped face images. The results are reported over the average of 5 runs over the same imbalanced dataset. (The imbalance statistics for all datasets are in the supplementary).

**Baselines and evaluation metrics** We design different baselines to show the importance of different components of our approach. **(i)** *Uniform InfoGAN* [10]: This is the original InfoGAN with fixed and uniform categorical distribution. **(ii)** *Ground-truth InfoGAN*: This is InfoGAN with a fixed, but imbalanced categorical distribution where the class probabilities reflect the ground-truth class imbalance. **(iii)** *Ground-truth InfoGAN + $L_{ntxent}$*: Similar to the previous baseline but with the contrastive loss (Eq. 3). **(iv)** *Gumbel-softmax*: Similar to InfoGAN, but this baseline does not have a fixed prior for the latent variables. Instead, the priors are learned using the Gumbel-softmax technique [30]. **(v)** *Gumbel-softmax + pos-$L_{ntxent}$*: This is the version where apart from having a learnable prior, we also apply a part of $L_{ntxent}$, where we enforce the positive pairs to have similar latent prediction $Q(c|x)$ but do not use negative pairs. **(vi)** *Elastic-InfoGAN*: This is our final model in which we use the complete form of $L_{ntxent}$. **(vii)** *JointVAE* [15]: We also include this VAE based baseline, which performs joint modeling of disentangled discrete and continuous factors. The objective function uses two KL-divergence loss terms to enforce the inferred discrete and continuous

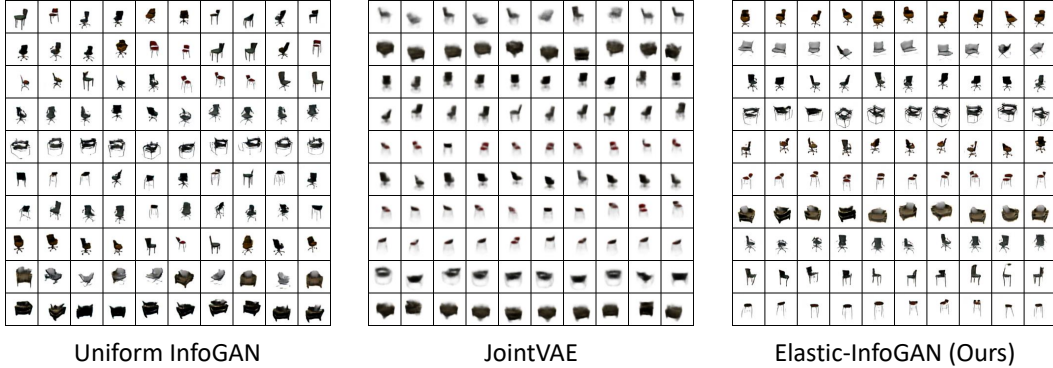

| Uniform InfoGAN | JointVAE | Elastic-InfoGAN (Ours) |

Figure 5: Image generations on a random imbalanced 3D Chairs split, on a randomly chosen sets of categories. Each row corresponds to a learned latent variable. Similar to the results on existing datasets, images are much more consistent corresponding to a latent variable for Elastic-InfoGAN, compared with Uniform-InfoGAN or JointVAE. Due to lack of details in the results for JointVAE, it is hard to figure out if some categories repeat in multiple rows.

latent variables to follow their respective prior distributions (e.g., uniform categorical and standard normal respectively). We tune the weights for both of these loss functions separately, and report the best configuration's results (see supplementary for more details).

The standard metrics for evaluating disentanglement require access to the ground-truth latent factors [16, 32, 24], which is rarely the case with real-world datasets. Furthermore, our evaluation should specifically capture the ability to disentangle *class-specific* information from other factors in an imbalanced dataset. Since the aforementioned metrics don't capture this property, we propose to use the following metrics: **(a) *Average Entropy (ENT)*:** Evaluates two properties: (i) whether the images generated for a given categorical code belong to the same ground-truth class i.e., whether the ground-truth class histogram for images generated for each categorical code has low entropy; (ii) whether each ground-truth class is associated with a single unique categorical code. We generate 1000 images for each of the $k$ latent categorical codes, compute class histograms using a pre-trained classifier[2] to get a $k \times k$ matrix (where rows index latent categories and columns index ground-truth categories). We report the average entropy across the rows (tests (i)) and columns (tests (ii)). **(b) *Normalized Mutual Information (NMI)*** [56]: We treat our latent category assignments of the fake images (we generate 1000 fake images for each categorical code) as one clustering, and the category assignments of the fake images by the pre-trained classifier as another clustering. NMI measures the correlation between the two clusterings. The value of NMI will vary between 0 to 1; higher the NMI, stronger the correlation.

**Implementation details** Transformations ($\delta$) used: (i) MNIST: Rotation ($\pm 10$ deg) + Zoom ($\pm 0.1\times$); for (ii) 3D Cars, (iii) 3D Chairs, and (iv) ShapeNet: Rotation ($\pm 10$ deg) + Random horizontal flip + Random crop (preserving 95% of image); (v) YouTube-Faces: Random horizontal flip + Random crop (scale image by $1.1\times$ and crop $64 \times 64$ patch) + Gamma contrast (gamma $\sim U(0.3, 4.0)$). Refer to supplementary for more implementation details.

## 4.1 Quantitative evaluation

**Comparisons to baselines assuming uniform prior** As explained in previous sections, baselines using this prior for the latent categorical distribution would (in theory) have difficulty in disentangling object identity as a separate factor in class imbalanced data. We observe this behavior empirically too (see Table 1); Elastic-InfoGAN, which learns the latent categorical distribution, obtains significant boosts of 0.113 and 0.127 in NMI, and -0.244 and -0.395 in ENT compared to the Uniform InfoGAN baseline for MNIST and YouTube-Faces, respectively. The boost is even more significant when compared to JointVAE: 0.209, 0.345 in NMI, and -0.4877, -0.1.081 in ENT for MNIST and YouTube-

|  | MNIST | | YTF | | 3D-cars | | 3D-chairs | | ShapeNet | |
| --- | --- | --- | --- | --- | --- | --- | --- | --- | --- | --- |
|  | NMI | ENT | NMI | ENT | NMI | ENT | NMI | ENT | NMI | ENT |
| JointVAE [15] | 0.704 | 0.661 | 0.485 | 1.554 | 0.458 | 1.024 | 0.480 | 1.817 | 0.189 | 1.101 |
| Uniform-InfoGAN [10] | 0.777 | 0.457 | 0.666 | 1.031 | 0.499 | 1.108 | 0.253 | 1.663 | 0.638 | 0.531 |
| Gumbel-Softmax | 0.836 | 0.326 | 0.760 | 0.757 | 0.400 | 1.318 | 0.236 | 1.696 | 0.603 | 0.619 |
| Gumbel-Softmax + pos-$L_{ntxent}$ | 0.878 | 0.235 | 0.765 | 0.719 | 0.582 | 0.919 | 0.454 | 1.209 | 0.724 | 0.397 |
| Elastic-InfoGAN (Ours) | **0.889** | **0.213** | **0.792** | **0.636** | **0.850** | **0.303** | **0.650** | **0.765** | **0.790** | **0.297** |
| Ground-truth InfoGAN | 0.783 | 0.412 | 0.694 | 0.961 | 0.451 | 1.191 | 0.174 | 1.837 | 0.549 | 0.673 |
| Ground-truth InfoGAN + $L_{ntxent}$ | 0.801 | 0.369 | 0.742 | 0.767 | 0.784 | 0.437 | 0.592 | 0.885 | 0.531 | 0.716 |

Table 1: Distentanglement quality measured by NMI (↑) and ENT (↓). The first five methods have no knowledge of the ground-truth distribution, while the last two methods do. We see that incorporating contrastive loss within a baseline (either Gumbel-Softmax or Ground-truth InfoGAN) helps the model better learn the disentangled representations. Overall, Elastic-InfoGAN demonstrates the ability to better disentangle object identity from other factors compared to baselines. (See supplementary for error bars.)

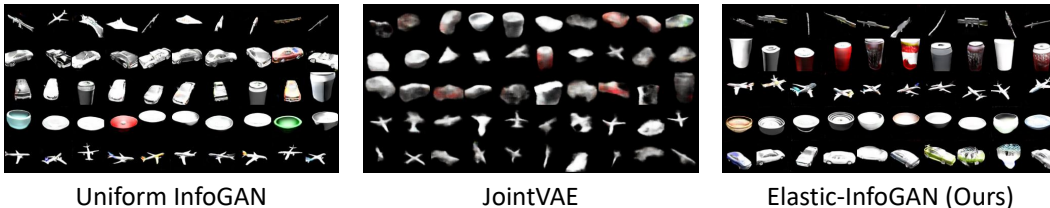

Uniform InfoGAN                JointVAE                Elastic-InfoGAN (Ours)

Figure 6: Image generations on ShapeNet: different categorical latent codes capture object identity (e.g., planes/cars) much more consistently in Elastic-InfoGAN than Uniform-InfoGAN, which mixes up generations (e.g., cars with cans in row 3). JointVAE shares this issue, in addition to poor quality of generations.

Faces, respectively. Similar results can be observed for Cars, Chairs, and ShapeNet, where our method gets a boost of 0.351, 0.397, 0.152 respectively in NMI, and -0.805, -0.897, -0.234 respectively in ENT, compared to Uniform InfoGAN.

**Effect of using contrastive loss** From Table 1, we see that the performance of baselines learning the prior distribution follow a particular order across all the datasets. Having an auxiliary constraint to enforce positive pairs to have similar representations improves performance (Gumbel-Softmax vs. Gumbel-Softmax + pos-$L_{ntxent}$), and additionally constraining the negative pairs to have dissimilar representations results in further gains (Gumbel-Softmax + pos-$L_{ntxent}$ vs. Elastic-InfoGAN). This trend indicates that the absence of appropriate auxiliary constraints results in the model having no signal to be invariant to undesirable factors. For example, one of the ways in which different 'ones' in MNIST vary is rotation, which can be used as a factor (as opposed to object identity) to group data in imbalanced cases (recall the different ways to group from Sec. 3.2). Similarly in Cars, if the number of different poses actually match the number of discrete categories, pose could emerge as a factor instead of object identity. These are the potential scenarios where Gumbel-Softmax/Ground-truth InfoGAN will perform more poorly than Elastic-InfoGAN (see supplementary for more results on this). Note that the transformations ($\delta$) used in $L_{ntxent}$ are not supposed to capture all the intra-class variations themselves; their role is to help the model ($Q$) focus more on object identity while predicting the latent category, by ruling out the variations in $\delta$ as a way to group.

Interestingly, using a fixed ground-truth prior (Ground-truth InfoGAN) does not always result in better disentanglement than learning the prior (Gumbel-softmax). This requires further investigation, but we hypothesis an explanation based on the idea presented in [11]. The *effective number of samples* in a category will not necessarily be the same as the number of instances in that category; e.g., two 0 digits that are almost equivalent to each other would result in an effective sample size closer to one instance rather than two instances. It is therefore possible that the distribution for the effective samples might not exactly match the ground-truth imbalanced distribution of instances, and hence using the ground-truth distribution for the latent space might result in sub-optimal disentanglement.

### 4.2 Qualitative evaluation

We next qualitatively evaluate our method's disentanglement ability. Figs. 3-7 show results for MNIST, Cars, Chairs, ShapeNet, and YouTube-Faces. Overall, Elastic-InfoGAN generates more

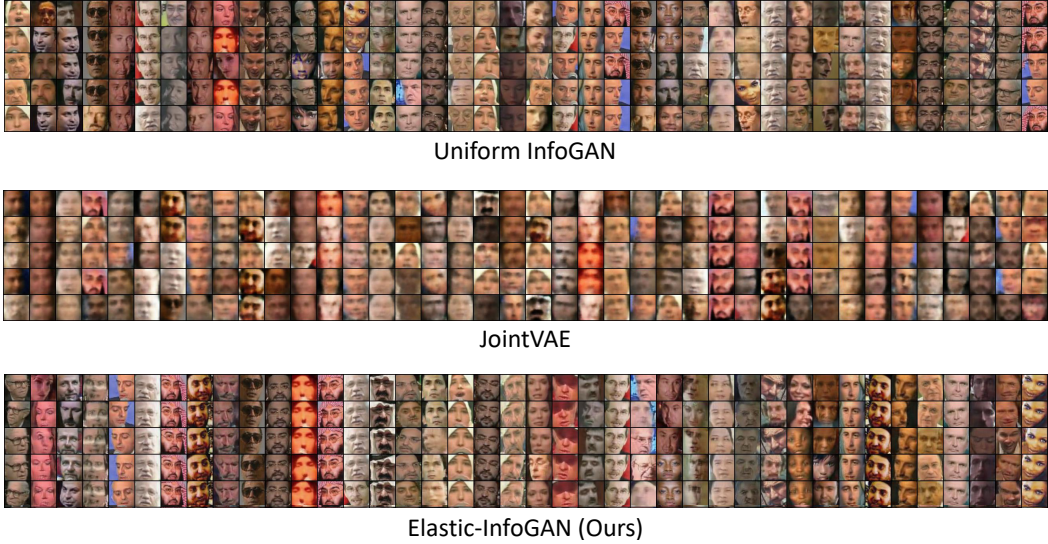

Uniform InfoGAN

JointVAE

Elastic-InfoGAN (Ours)

Figure 7: Image generations on YouTube-Faces. Each column corresponds to a latent variable. Although there are a few redundant latent variables (e.g., 7th and 13th columns) in Elastic-InfoGAN, it generates images belonging to the same person more consistently compared to Uniform-InfoGAN and JointVAE, which tend to mix up face identities a lot more frequently.

consistent images for each latent code compared to Uniform InfoGAN and JointVAE. For example, in Fig. 3, our model only generates inconsistent images in the 4th row (mixing up 8 with 3's) whereas the baselines generate inconsistent images in several rows. In Fig. 4 and Fig. 5, we see that for a given latent variable, our model can consistently generate images from the same object category, in different pose/viewpoints. For Chairs, there are some cases in which the images generated in different rows look similar for Elastic-InfoGAN, but this still happens much less frequently than Uniform InfoGAN and JointVAE. Fig. 6 further demonstrates the ability of Elastic-InfoGAN to become invariant to other forms of continuous factors (e.g. object pose), with different categorical codes accurately representing different high-level object categories; e.g,. rifles vs cars, in a better way than the baseline methods. Similarly, in Fig. 7, our model generates faces of the same person corresponding to a latent variable more consistently than the baselines. Both Uniform InfoGAN and JointVAE, on the other hand, more often mix up identities within the same categorical code because they incorrectly assume a prior uniform distribution.

**Modeling continuous factors**  Finally, we demonstrate that Elastic-InfoGAN does not impede modeling of continuous factors in the imbalanced setting. Specifically, one can augment the input with continuous codes (e.g., r1, $r2 \sim U$ (-1, 1)) along with the existing categorical and noise vectors. In the right figure, we show the results of continuous code interpolation; we can see that each of the 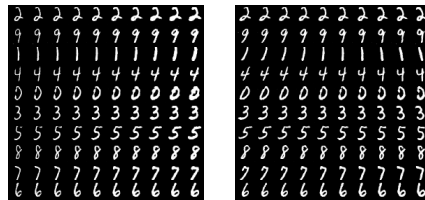 continuous codes largely captures a particular continuous factor (stroke width on left, digit rotation on right).

## 5    Conclusion

We proposed an unsupervised generative model that better disentangles object identity as a factor of variation, without the knowledge of class imbalance. Although there are some limitations (e.g., its applicability in highly skewed data), we believe that we address an important, unexplored problem setting. Similar to how the area of supervised machine learning has evolved over time to account for class imbalance in real world data, our hope with this work is to pave the way for the evolution of unsupervised learning based methods to work well in class imbalanced data, which is inevitable if these algorithms have to be deployed in the real world.

## Broader Impact

Image datasets, particularly the ones with human faces, have a potential problem of not being diverse enough. This conceptualizes in some form of imbalance in the dataset, where, for example, a dataset of human faces might not represent faces from all ethnical communities in appropriate proportions. A popular application of GANs is to use synthetic images for data augmentations. With traditional GANs, however, it is possible that the underrepresented classes might not be modeled as accurately (mode dropping problem), thus limiting their applicability. The idea presented in this work is specifically tailored to handle such cases, by discovering both, the over and under-represented classes. This could then enable data augmentation using generated images, and help increase the proportions of underrepresented classes.

GANs in general pose some ethical concerns, in terms of creating/altering visual content (e.g., deepfakes). Our work, which is a derivative of GAN, is no exception in that regard, as it could have some malicious applications, such as image fabrication. We do want to point out that such applications are not that straightforward with the method proposed in its current form, as our method doesn't operate directly on real images (the input to the generator is latent vectors).

## Acknowledgments and Disclosure of Funding

This work was supported in part by NSF IIS-1751206, IIS-1748387, IIS-1812850, IIS-1901527, IIS-2008173, AWS ML Research Award, Google Cloud Platform research credits, and Adobe Data Science Research Award.

## Footnotes

[1]we use - `https://github.com/panmari/stanford-shapenet-renderer`

[2]We train the classifier by creating a 80/20 train/val split on a per class basis. Classification accuracies: (i) MNIST: 98%, (ii) 3D Cars: 99%, (iii) 3D Chairs: 97%, (iv) ShapeNet: 95%, (v) YouTube-Faces: 96%. See supplementary for details.

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
