[Supplementary Material]

# Supplementary Material:
# Elastic-InfoGAN: Unsupervised Disentangled Representation Learning in Class-Imbalanced Data

**Utkarsh Ojha**[1]     **Krishna Kumar Singh**[1,2]     **Cho-Jui Hsieh**[3]     **Yong Jae Lee**[1]

[1]UC Davis          [2]Adobe Research          [3]UCLA

`utkarshojha.github.io/elastic-infogan/`

In this document, we first continue our discussion of the importance of constrastive learning of representations to disentangle object identity. We then mathematically analyze the applicability of contrastive loss in a generic class-imbalanced setting, studying how appropriate our assumptions pertaining to $L_{ntxent}$ are. We then demonstrate an application of our framework, making use of the learned inference network for nearest neighbor classification. Finally, we discuss all the implementation details, and provide detailed information about the random imbalanced splits used for different datasets.

## 1 Importance of constrastive loss

In this section, we continue the discussion initiated in Sec. 4.4 of the main paper, about the importance of $L_{ntxent}$. Specifically, we discussed the potential cases where Ground-truth and Gumbel-Softmax InfoGAN baselines (which could in theory produce accurate disentanglement) would perform poorly. In Fig.1, we visualize these scenarios qualitatively. We highlight the cases where the mentioned baselines produce undesirable groupings; i.e., generating cars having *similar* pose but *different* identity for the same latent code. Our approach, for the same split of random classes, produces groupings which are much more coherent; i.e., generating cars of same identity rendered in different positions, for a latent code.

Finally, Table 1 and 2 cover the quantitative results presented in Table 1 of the main paper with error bars (standard deviation). In general, the error bars for our method are lower than the baselines, indicating more robustness is introduced via $L_{ntxent}$. The numbers are over (i) 50 different runs for MNIST (corresponding to 50 imbalanced splits), (ii) 25 runs for 3d-Cars/Chairs (refer to the main paper about the creation of 25 splits), and (iii) 5 runs over the same imbalanced split for YouTube Faces.

## 2 Analysing the constrastive loss in imbalanced scenarios

As mentioned in Sec. 3.2 of the main paper, we approximate negative samples with randomly sampled pairs $I_i$, $I_j$ ($i \neq j$) in a batch while using $L_{ntxent}$. Since we don't know the labels during training, some of these sampled pairs could turn out to be *false negatives* (images belonging to the same category). In this section, we quantify the extent to which these false negatives impact the training of overall model.

Assume that the proportion of different classes in the dataset is denoted by $p_1, p_2, p_3, ..., p_k$ (where $k$ denotes the number of classes, and $p_i$ denotes $i^{th}$ class probability). Let's say that in a batch, the algorithm constructs $N$ negative pairs, where each pair is equally likely to be a false negative.

| | Ground-truth InfoGAN | Gumbel-Softmax InfoGAN | Ours |

Figure 1: Image generations on a random imbalanced 3D Cars split. Baselines which could potentially capture object identity as a factor of variation, but lack $L_{ntxent}$, sometimes generate undesirable groupings, e.g. focusing on pose/orientation rather than object identity (groups highlighted with an outline: *red* for Ground-truth InfoGAN, *blue* for Gumbel-Softmax InfoGAN). Our approach, for the same set of object categories, produces better groupings focusing on identity and generating cars with varying pose/orientation.

| | MNIST | YTF | Cars | Chairs | ShapeNet |
|---|---|---|---|---|---|
| JointVAE [2] | $0.6801 \pm 0.081$ | $0.4472 \pm 0.027$ | $0.3915 \pm 0.236$ | $0.4478 \pm 0.212$ | $0.1892 \pm 0.075$ |
| Uniform InfoGAN [1] | $0.7765 \pm 0.045$ | $0.6656 \pm 0.005$ | $0.4990 \pm 0.205$ | $0.2525 \pm 0.165$ | $0.6382 \pm 0.088$ |
| Gumbel-softmax | $0.8360 \pm 0.048$ | $0.7603 \pm 0.014$ | $0.4003 \pm 0.309$ | $0.2356 \pm 0.134$ | $0.6031 \pm 0.153$ |
| Gumbel-softmax + pos-$L_{ntxent}$ | $0.8781 \pm 0.061$ | $0.7647 \pm 0.011$ | $0.5818 \pm 0.127$ | $0.4536 \pm 0.172$ | $0.7236 \pm 0.062$ |
| Elastic-InfoGAN (Ours) | $\mathbf{0.8893 \pm 0.044}$ | $\mathbf{0.7923 \pm 0.004}$ | $\mathbf{0.8504 \pm 0.068}$ | $\mathbf{0.6499 \pm 0.082}$ | $\mathbf{0.7900 \pm 0.077}$ |
| Ground-truth InfoGAN | $0.7827 \pm 0.049$ | $0.6941 \pm 0.031$ | $0.4512 \pm 0.276$ | $0.1738 \pm 0.142$ | $0.5492 \pm 0.152$ |
| Ground-truth InfoGAN + $L_{ntxent}$ | $0.8008 \pm 0.052$ | $0.7421 \pm 0.023$ | $0.7841 \pm 0.092$ | $0.5915 \pm 0.152$ | $0.5314 \pm 0.105$ |

Table 1: Distentanglement quality measured by NMI ($\uparrow$ better). The first five methods do not have knowledge of the ground-truth distribution, while the last two methods do. Our model outperforms the baselines with/without the knowledge of ground-truth class distribution, for all datasets, with relatively low error-bar

Consider one such pair, $I_i$ and $I_j$: the probability that this pair will be a false negative (F.N.) is $P(F.N.) = \sum_{i=1}^{k} p_i^2$

Since all pairs are sampled independently, the expected fraction of false negatives, $\mathbf{E}(F.N.)$ in the batch hence becomes:

$$\frac{N \times \sum_{i=1}^{k} p_i^2}{N} = \sum_{i=1}^{k} p_i^2 \qquad (1)$$

We wish for this value to be as small as possible. As expected, the minimum value is attained when the dataset is balanced, i.e. $p_1 = p_2... = p_k = \frac{1}{k}$, where $\mathbf{E}(F.N.)$ becomes $\frac{1}{k}$. The maximum value, on the other hand, is attained when one class completely dominates the other classes ($p_i = 1$ for some $i$, and $p_j = 0$ ($\forall j \neq i$)), making $\mathbf{E}(F.N.) = 1$. So, as long as there are even a few prominent classes (among $k$), the expected false negatives remain considerable low. For instance, consider the randomly created imbalanced splits for MNIST (total classes = 10; detailed in Sec.5.0.1): the average $\mathbf{E}(F.N.)$ for the 50 splits is $0.119 \pm 0.008$. This means that on average, 88% of the negative pairs will be true negative, which is only 2% less than the ideal 90% (when $\mathbf{E}(F.N.) = \frac{1}{k}$). We further verified this hypothesis for the real-world imbalanced dataset used in this work, YouTube Faces (total classes = 40; split detailed in Sec.5.0.5). The $\mathbf{E}(F.N.)$ for this dataset is 0.0363, which means 96% of negative pairs are true negatives (only 1% less than ideal scenario).

We want to emphasize the key take away from this analysis: it is *not* that the datasets explored in this work don't have sufficient imbalance (the reader can verify this by looking at the actual splits described in Sec.5.0.1 - Sec.5.0.5), but rather that the mathematical formulation of $\mathbf{E}(F.N.)$ happens to allow a large spectrum of imbalanced datasets to be applicable while using $L_{ntxent}$.

|  | MNIST | YTF | Cars | Chairs | ShapeNet |
|---|---|---|---|---|---|
| JointVAE [2] | $0.7006 \pm 0.134$ | $1.7173 \pm 0.022$ | $1.0818 \pm 0.526$ | $1.9090 \pm 0.438$ | $1.0113 \pm 0.064$ |
| Uniform InfoGAN [1] | $0.4569 \pm 0.096$ | $1.0312 \pm 0.009$ | $1.1075 \pm 0.483$ | $1.6626 \pm 0.368$ | $0.5318 \pm 0.141$ |
| Gumbel-softmax | $0.3260 \pm 0.101$ | $0.7573 \pm 0.014$ | $1.3183 \pm 0.702$ | $1.6960 \pm 0.305$ | $0.6188 \pm 0.241$ |
| Gumbel-softmax + pos-$L_{ntxent}$ | $0.2347 \pm 0.101$ | $0.7188 \pm 0.006$ | $0.9188 \pm 0.247$ | $1.2088 \pm 0.417$ | $0.3973 \pm 0.069$ |
| Elastic-InfoGAN (Ours) | $\mathbf{0.2130 \pm 0.088}$ | $\mathbf{0.6358 \pm 0.015}$ | $\mathbf{0.3026 \pm 0.147}$ | $\mathbf{0.7651 \pm 0.180}$ | $\mathbf{0.2972 \pm 0.104}$ |
| Ground-truth InfoGAN | $0.4196 \pm 0.097$ | $0.9611 \pm 0.009$ | $1.1907 \pm 0.646$ | $1.8369 \pm 0.318$ | $0.6730 \pm 0.249$ |
| Ground-truth InfoGAN + $L_{ntxent}$ | $0.3694 \pm 0.094$ | $0.7672 \pm 0.007$ | $0.4373 \pm 0.191$ | $0.8851 \pm 0.331$ | $0.7162 \pm 0.172$ |

Table 2: Distentanglement quality measured by ENT ($\downarrow$ better).

|  | MNIST | YTF | 3D Cars | 3D Chairs |
|---|---|---|---|---|
| Randomly initialized | $0.4865 \pm 0.009$ | $0.9541 \pm 0.003$ | $0.3140 \pm 0.055$ | $0.2942 \pm 0.038$ |
| Uniform InfoGAN [1] | $0.8665 \pm 0.014$ | $0.9581 \pm 0.026$ | $0.7700 \pm 0.074$ | $0.6438 \pm 0.109$ |
| Gumbel-Softmax | $0.9113 \pm 0.017$ | $0.9812 \pm 0.013$ | $0.6726 \pm 0.073$ | $0.6241 \pm 0.119$ |
| Elastic-InfoGAN (Ours) | $\mathbf{0.9655 \pm 0.007}$ | $\mathbf{0.9985 \pm 0.018}$ | $\mathbf{0.9852 \pm 0.005}$ | $\mathbf{0.9515 \pm 0.020}$ |

Table 3: 1NN classification accuracy (%) of different baselines. By learning to better disentangle object identity from other factors, our method can infer much better representations needed for the task of nearest neighbor classification.

# 3  k-Nearest neighbour classification

We discussed in Sec. 1 and 3.2 of the main paper that disentangling object identity from other factors can have downstream applications. One such application can be demonstrated via the trained inference network ($Q$), where it can be used to extract features pertaining to object identity of any real image $x$. We present results in scenarios of class-imbalance, where it becomes difficult to perform grouping based on object identity.

Table 3 summarizes the performance of different methods for the task of nearest neighbours classification. Randomly initialized refers to using the same inference network architecture with random weights. In particular, we create a 80/20 (train/test) split of real images, and report classification accuracy when images from the test split are used as queries. We see that our method consistently achieves superior performance across the four datasets. Note that the performance on YTF is remarkably good because images for each class are almost identical, so the embedding for image for each class will be very close to each other, regardless of the encoder.

Fig. 2 illustrates some sample queries, and the corresponding nearest neighbors retrieved using uniform InfoGAN and our method. The neighbors extracted using uniform InfoGAN suffer inconsistency, where sometimes the nearest neighbor has similarities in pose (3rd row), rough color (4th row) etc. Our method, on the other hand, has much more success in retrieving images belonging to the same category. Furthermore, note the variations among the query image and the extracted images for our method. It is not possible to cover all these differences in pose, azimuth through simple transformations ($\delta$) that we use in $L_{ntxent}$. This is an indication that our method is not simply memorizing the transformations that we introduce, and is actually learning representations which focus on object identity in general, leading to superior performance in 1NN classification.

# 4  Implementation details (continued)

## 4.1  MNIST

For MNIST, we operate on the original 28x28 images, with a 10-dimensional categorical code to represent the 10 digit categories, and 62 noise variables sampled from a normal distribution. We follow the exact architecture as described in InfoGAN [1]: The generator network $G$ takes as input a 64 dimensional noise vector $z \sim \mathcal{N}(0, 1)$ and 10 dimensional samples from a Gumbel-Softmax distribution. The discriminator $D$ and the latent code prediction network $Q$ share most of the layers except the final fully connected layers.

The pre-trained classification architecture used for evaluation for MNIST consists of 2 Conv + 2 FC layers, with max pool and ReLU after every convolutional layer.

Figure 2: Nearest neighbors obtained using our method vs. uniform InfoGAN. We see that our method retrieves diverse cars (in terms of azimuth, pose etc.) belonging to the same category. Neighbors extracted using Uniform InfoGAN, on the other hand, suffer in terms of categorical consistency.

## 4.2 3D Cars and Chairs

We follow identical steps for 3D Cars and Chairs. For 3D Cars, we follow the procedure explained in the main paper to select 10 categories. We resize all the renderings to 64x64 resolution, and use a 10 dimensional categorical code to represent 10 object identities, and use 100 noise variables to capture other variations (pose/viewpoints etc.).

Our architecture is based on the one proposed in StackGANv2 [4], where we use its 1-stage version for generating 64x64x3 resolution images. There is an initial fully connected layer which maps the input (concatenation of $z$ and $c$) to an intermediate feature representation. A series of a combination of upsampling + convolutional layers (interleaved with batch normalization and Gated Linear Units) increase the spatial resolution of the feature representation, starting from 1024 (feature size: 4 x 4 x 1024) channels to 64 (feature size: 64 x 64 x 64) channels. A convolutional network transforms the feature representation into a 3 channel output, while maintaining the spatial resolution, which serves as the fake image. The discriminator network consists of 4 convolutional layers interleaved with batch normalization and leaky ReLU layers, which serve as the common layers for both the $D$ and $Q$ networks. After that, $D$ has one non-shared convolutional layer, which maps the feature representation into a scalar value reflecting the real/fake score. For $Q$, we have a pair of non-shared convolutional layers which map the feature representation into a 10 dimensional latent code prediction.

The classifier used for evaluating results on 3D Cars/Chairs is a ResNet-50 network, trained on the complete data with a 80/20 train/validation split (different pre-trained networks for different sets of 10 classes chosen).

### 4.3 YouTube-Faces

For YouTube-Faces, we crop the faces using the provided bounding box annotations, and then resize them to 64x64 resolution, and use a 40-dimensional categorical code to represent 40 face identities (first 40 categories sorted in alphabetical manner, as done in [3]), and 100 noise variables.

The architecture for the Generator/Discriminator is very much similar to that used for 3D Cars/Chairs, except that we have one more stage, which takes in the 64 x 64 x 64 resolution intermediate features and translates that into another fake image. We apply our losses on images from both the stages, and the images generated from the second stage are used for evaluation purposes.

The classifier used for evaluating results on YouTube-Faces is also ResNet-50 network (similar to the one used for 3D Cars/Chairs), but we pretrain it on VGGFace2, before fine-tuning on YouTube-Faces.

### 4.4 Training details

We employ a similar way of training the generative and discriminative modules as described in [1]. We first update the discriminator based on the real/fake adversarial loss. In the next step, after computing the remaining losses (mutual information + $L_{ntxent}$), we update the generator ($G$) + latent code predictor ($Q$) + latent distribution parameters ($p_{i_s}$) at once. Our optimization process alternates between these two phases. $L_{ntxent}$ is computed on features obtained from penultimate layer, after the leaky ReLU activation. One good reason to not use ReLU instead is that it will produce bias for cosine similarity distance, since all the feature values will be positive. We use Adam optimizer, with a learning rate of 0.0002. For MNIST, we train all baselines for 200 epochs, with a batch size of 64. For 3D Cars/Chairs, we train for 600 epochs, with a batch size of 50. For YouTube-Faces, we train until convergence, as measured via qualitative realism of the generated images, using a batch size of 50. $\lambda_2$ in $L_{final}$ is set to 10 to balance the magnitude of the different loss terms. $\tau = 0.1$ is used for sampling from Gumbel-Softmax, which results in samples having very low entropy (very close to one hot vectors from a categorical distribution).

**JointVAE details:** We use the KL term for both continuous as well as discrete variables, to follow the standard normal ($\mathcal{N}(0, 1)$) and uniform categorical distribution ($Cat(p = 1/k)$), respectively. We use uniform categorical because of the unsupervised nature of the problem (L53-5, 83-4). We use the same weight ($\beta$) for both KL loss terms (similar to JointVAE paper), the value of which was first decided empirically based on image reconstruction quality - we observed that a value in 100s (e.g. 100-300) resulted in poor reconstruction quality. After that, we report the results for best performing model by ablating $\beta_{cont}$ and $\beta_{disc}$ from the set $\{10, 20, 30, 40, 50\}$.

Finally, one behavior we observe is that if the random initialization of class probabilities is too skewed (only few classes have high probability values), then it becomes very difficult for them to get optimized to the ideal state. We hence initialize them with the uniform distribution, which makes training much more stable.

Experiments on MNIST, which involve running 50 versions, can be done in parallel across 4 NVIDIA Tesla V100 GPUs (16 GB RAM). On each of them, around 6 (out of 50) can run in parallel. In this manner, it takes about 8 hours to complete the training on 50 MNIST imbalanced splits. For 3D Cars/Chairs, it takes about 4-5 hours if proper parallelism is employed on the same system. For YTF, it takes about 4 hours to run 5 versions on the same imbalanced split (while running in parallel).

## 5 Ground truth class imbalance

Here we describe the exact class imbalance used in our experiments. For MNIST, we include below the 50 random imbalances created. For 3D Cars/Chairs, we first describe the class ids of the randomly chosen 10 categories for all the 5 sets, then we describe the 5 random imbalanced splits used on each of these sets. For YouTube-Faces, we include the true ground truth class imbalance in the first 40 categories. The imbalances reflect the class frequency.

### 5.0.1 MNIST

- 0.147, 0.037, 0.033, 0.143, 0.136, 0.114, 0.057, 0.112, 0.143, 0.078
- 0.061, 0.152, 0.025, 0.19, 0.12, 0.036, 0.092, 0.185, 0.075, 0.064
- 0.173, 0.09, 0.109, 0.145, 0.056, 0.114, 0.075, 0.03, 0.093, 0.116

- 0.079, 0.061, 0.033, 0.139, 0.145, 0.135, 0.057, 0.062, 0.169, 0.121
- 0.053, 0.028, 0.111, 0.142, 0.13, 0.121, 0.107, 0.066, 0.125, 0.118
- 0.072, 0.148, 0.092, 0.081, 0.119, 0.172, 0.05, 0.109, 0.085, 0.073
- 0.084, 0.143, 0.07, 0.082, 0.059, 0.163, 0.156, 0.063, 0.074, 0.105
- 0.062, 0.073, 0.065, 0.183, 0.099, 0.08, 0.05, 0.16, 0.052, 0.177
- 0.139, 0.113, 0.074, 0.06, 0.068, 0.133, 0.142, 0.13, 0.112, 0.03
- 0.046, 0.128, 0.059, 0.112, 0.135, 0.164, 0.142, 0.125, 0.051, 0.037
- 0.107, 0.057, 0.154, 0.122, 0.05, 0.111, 0.032, 0.044, 0.136, 0.187
- 0.129, 0.1, 0.039, 0.112, 0.119, 0.095, 0.047, 0.14, 0.156, 0.064
- 0.146, 0.08, 0.06, 0.072, 0.051, 0.119, 0.176, 0.11, 0.158, 0.028
- 0.035, 0.051, 0.112, 0.143, 0.033, 0.165, 0.082, 0.165, 0.054, 0.161
- 0.041, 0.1, 0.073, 0.054, 0.155, 0.117, 0.091, 0.124, 0.142, 0.104
- 0.052, 0.139, 0.128, 0.133, 0.104, 0.107, 0.058, 0.137, 0.036, 0.107
- 0.055, 0.138, 0.059, 0.074, 0.08, 0.135, 0.085, 0.064, 0.172, 0.139
- 0.141, 0.156, 0.119, 0.062, 0.08, 0.022, 0.043, 0.159, 0.101, 0.118
- 0.11, 0.088, 0.033, 0.062, 0.089, 0.176, 0.161, 0.105, 0.144, 0.032
- 0.157, 0.111, 0.125, 0.099, 0.036, 0.119, 0.036, 0.05, 0.147, 0.121
- 0.119, 0.121, 0.117, 0.152, 0.026, 0.174, 0.027, 0.065, 0.151, 0.049
- 0.057, 0.07, 0.134, 0.118, 0.058, 0.185, 0.07, 0.13, 0.116, 0.063
- 0.102, 0.082, 0.135, 0.046, 0.128, 0.106, 0.116, 0.085, 0.133, 0.066
- 0.057, 0.193, 0.2, 0.123, 0.022, 0.154, 0.115, 0.025, 0.065, 0.047
- 0.056, 0.196, 0.168, 0.052, 0.116, 0.062, 0.099, 0.133, 0.065, 0.053
- 0.04, 0.022, 0.2, 0.194, 0.038, 0.033, 0.161, 0.097, 0.159, 0.056
- 0.04, 0.036, 0.119, 0.204, 0.16, 0.103, 0.089, 0.061, 0.136, 0.052
- 0.112, 0.189, 0.145, 0.163, 0.113, 0.031, 0.028, 0.062, 0.045, 0.112
- 0.071, 0.099, 0.113, 0.175, 0.082, 0.068, 0.03, 0.066, 0.133, 0.164
- 0.134, 0.074, 0.111, 0.091, 0.051, 0.119, 0.044, 0.085, 0.144, 0.148
- 0.103, 0.126, 0.084, 0.117, 0.084, 0.127, 0.131, 0.092, 0.117, 0.019
- 0.096, 0.121, 0.026, 0.046, 0.043, 0.124, 0.165, 0.04, 0.127, 0.213
- 0.117, 0.115, 0.125, 0.128, 0.081, 0.103, 0.073, 0.044, 0.137, 0.077
- 0.037, 0.021, 0.143, 0.165, 0.075, 0.111, 0.028, 0.132, 0.134, 0.154
- 0.154, 0.049, 0.128, 0.089, 0.082, 0.072, 0.034, 0.138, 0.108, 0.146
- 0.078, 0.141, 0.084, 0.139, 0.085, 0.062, 0.035, 0.174, 0.15, 0.053
- 0.112, 0.112, 0.128, 0.112, 0.107, 0.142, 0.032, 0.142, 0.063, 0.049
- 0.084, 0.091, 0.128, 0.129, 0.045, 0.105, 0.05, 0.091, 0.089, 0.188
- 0.062, 0.136, 0.112, 0.153, 0.091, 0.046, 0.089, 0.03, 0.161, 0.12
- 0.143, 0.1, 0.046, 0.166, 0.107, 0.191, 0.026, 0.078, 0.097, 0.047
- 0.077, 0.174, 0.05, 0.098, 0.028, 0.173, 0.067, 0.106, 0.096, 0.13
- 0.105, 0.022, 0.183, 0.056, 0.045, 0.103, 0.081, 0.135, 0.119, 0.149
- 0.083, 0.127, 0.126, 0.028, 0.209, 0.03, 0.066, 0.125, 0.1, 0.107
- 0.138, 0.142, 0.074, 0.091, 0.103, 0.067, 0.12, 0.04, 0.1, 0.124
- 0.058, 0.039, 0.088, 0.113, 0.093, 0.055, 0.162, 0.069, 0.168, 0.155
- 0.02, 0.162, 0.133, 0.138, 0.137, 0.051, 0.069, 0.032, 0.118, 0.14
- 0.071, 0.046, 0.134, 0.119, 0.159, 0.057, 0.039, 0.135, 0.057, 0.184

### 5.0.2 3D Cars

List of class ids for different (total 5) sets of randomly chosen classes:

- 009, 002, 004, 007, 001, 025, 026, 024, 043, 023
- 096, 118, 040, 052, 024, 046, 123, 187, 150, 072
- 112, 019, 030, 037, 069, 056, 161, 193, 190, 061
- 038, 111, 104, 159, 035, 037, 086, 043, 173, 196
- 113, 009, 031, 016, 022, 078, 083, 060, 098, 100

Imbalances splits applied on each of these sets:

- 0.141, 0.116, 0.128, 0.077, 0.104
- 0.035, 0.137, 0.027, 0.068, 0.175
- 0.081, 0.076, 0.117, 0.109, 0.079
- 0.134, 0.108, 0.048, 0.143, 0.107
- 0.033, 0.111, 0.155, 0.160, 0.167

### 5.0.3 3D Chairs

List of class ids for different (total 5) sets of randomly chosen classes:

- 0965, 0960, 0710, 0045, 1332, 0996, 1074, 0236, 0098, 1196
- 0241, 0307, 0091, 1071, 1317, 0104, 1098, 1064, 0158, 0784
- 0565, 0326, 0892, 0308, 0858, 1212, 0802, 0236, 0257, 0749
- 0241, 0574, 0864, 0401, 1372, 1032, 1101, 0439, 0528, 0264
- 0561, 0334, 1036, 0724, 1314, 0766, 0572, 0840, 1338, 0991

Imbalances splits applied on each of these sets:

- 0.190, 0.040, 0.107, 0.170, 0.101
- 0.164, 0.204, 0.060, 0.164, 0.055
- 0.084, 0.119, 0.188, 0.067, 0.070
- 0.035, 0.178, 0.130, 0.102, 0.173
- 0.044, 0.060, 0.191, 0.100, 0.022

### 5.0.4 YouTube-Faces

- 0.019, 0.013, 0.024, 0.020, 0.028, 0.022, 0.053, 0.010, 0.062, 0.031, 0.037, 0.005, 0.011, 0.027, 0.034, 0.033, 0.009, 0.006, 0.011, 0.016, 0.024, 0.047, 0.028, 0.069, 0.012, 0.006, 0.024, 0.005, 0.006, 0.024, 0.005, 0.037, 0.028, 0.056, 0.059, 0.026, 0.008, 0.006, 0.028, 0.028

### 5.0.5 ShapeNet

- 0.1851, 0.1481, 0.1111, 0.2592, 0.2962

Results are reported averaged over 5 different runs on this imbalance split.