[Reviews · NeurIPS 2020]

Review 1

Summary and Contributions: The authors propose an extension of InfoGAN which addresses the issue of incoherent latent factors when trained on imbalanced datasets. They do this by proposing that the categorical distribution (which is typically uniform) is parameterised by learnable probabilities (via Gumbel-Softmax). Furthermore, in order to encourage that the latent groups corresponding to that distribution represent high-level features, a contrastive-based loss is used in one of the hidden layers of the auxiliary classifier network Q(c|x) similar to what is done in the recent SimCLR paper (Chen et al, 2020).

Strengths: - The issue is clearly presented and the proposed solution well-justified. - Decent quantitative evaluation (in some areas), and good qualitative evaluation.

Weaknesses: - While there is a decent empirical evaluation (since you evaluated on four datasets and two metrics), I feel that the baseline you have employed isn't strong enough and there are specific details missing, which makes me skeptical of some of your results. For example, for your JointVAE, I don't see mentioned any details regarding what values of beta were chosen for the KL term, and whether the KL term was employed for z (continuous vars) for some prior p(z), pi (discrete vars) for some prior p(pi), or both. There is an extensive literature (see FactorVAE, BetaVAE) on this and how it influences disentanglement in the latent codes, so it is relevant to compare against here. At the very least, it would have been nice to see one of the disentanglement metrics shown as a function of the strength of the KL term for pi. - "... JointVAE assume uniformly distributed data" --> well, yes and no -- it depends on what prior you choose and the strength of it. Even if you did impose a uniform prior Cat(1/K) on the discrete latent codes pi, this is just a prior distribution. Whether or not your inference model q(pi|x) (or aggregate posterior q(pi)) is close to that prior depends on the strength of the KL term you assign to it. As mentioned above as well, it is unclear how you dealt with this hyperparameter. - The Joint-VAE baseline indeed suffers from poor reconstruction quality, which is partly due to autoencoders in general but also in part due to the tradeoff between KL and reconstruction. Because of that, I would have preferred to see a stronger baseline that addresses this, like Factor-VAE.

Correctness: Claims are not strongly supported due to the lack of a strong baseline, but the method and methodology are reasonable.

Clarity: Yes.

Relation to Prior Work: Yes.

Reproducibility: Yes

Additional Feedback: - You mention averaging over runs in your results but I do not see standard deviation / variance in your table of results. I see it in your supplementary material, and you should add these into the main paper. -------- REVIEW UPDATE: I have read the fellow reviews as well as the rebuttal. The positive points for me are: - They expanded on Joint-VAE results with tuning beta and detailed the hps for the them, as well as report variances. I think they should have tuned the KL terms for both p(pi) and p(z) separately, but that would be a 2d grid search (i.e. more expensive) and I didn't necessarily expect that for the rebuttal. So I will suggest they do this in the final paper - They addressed another reviewer's concern with there being no InfoGAN+InfoNCE loss, and that has been added. - ShapeNet experiments are a nice addition. I think I am happy to bump this up to a 7 (from an initial score of 4). The new score however is made assuming at least some of the following suggestions are done in the final paper: - Tune the KLs for both continuous/discrete KLs individually, i.e. do a 2d grid search. - Do an analysis comparing GT class distributions with what is inferred by Gumbel-Softmax. You could use something like Wasserstein distance to compare the two (discrete) distributions. This might help you explore further why the ground truth distribution experiments are not performing as well. - For the disentanglement metrics, is average entropy (ENT) a new metric you have proposed, or has it been used in existing literature? If it is new, justify its use. I also think that you could use an appropriate metric (like FID) to potentially examine to what degree there is mode dropping between the methods compared. This could be good for the supplementary material. Thanks.


Review 2

Summary and Contributions: The authors point out the issue of uniform assumption in InfoGAN which works less effectively on imbalanced data. To address the issue, the work introduces the Gumbel-Softmax algorithm to learn parameterized class probabilities through back propagation, aiming to improve the generalization to imbalanced dataset without class distribution prior. Contrastive regularization is used for encode object identitiy into representation learning of the networks. The experiments on three artificial and a real-world imbalanced datasets validate the method.

Strengths: The authors address the issues in the scenario of imbalanced data distribution, which is an interesting and valuable perspective for generative tasks, as related researches of the topic typically focus on discriminative tasks. The use of Gumbel-Softmax algorithm enables a learnable class distribution, without the need of prior knowledge on the distribution. Integrating contrastive regularization with the unsupervised GAN framework facilitate encoding the factor of object identities into the representation learning. The comparison between "Gumbel-softmax + contrastive loss" and "Ground-truth + contrastive loss" show the benefit of the method on improving the performance of generative models.

Weaknesses: The most important question I'm concerned with is the relation between the issue of imbalanced distribution in the generative tasks and the effects of the two technical components, i.e., learning class distribution with Gumbel-Softmax and object identities with contrastive regularization. Intuitively, learnable class distribution can facilitate the proper adaption to real dataset with imbalanced distribution. While as the authors illustrated in the figure (Line 128- 134), the issue is inevitable and the performance gain achieved by the technique is less obvious, as shown in Table 1, the performance of Gumbel-softmax is inferior to the baseline of uniform distribution on 3D Cars and 3D Chairs. The first improvement on performance does not always hold across different datasets while the second improvement that learns object identities to benefit performance seems not closely related to the issue of imbalanced distribution. The technique is likely to benefit the uniform-InfoGAN (it would be better to provide the results of "Uniform + contrastive loss" for reference). Moreover, it is unclear if the advantage induced by the self-supervised technique can be replaced by using a simple condition GAN with class labels. The technical novelty and contribution of the paper are incremental. Both the Gumbel-Softmax trick and contrastive loss are proposed in the existing works and used in the work straightforwardly.

Correctness: Yes.

Clarity: The paper is well written and easy to follow.

Relation to Prior Work: The paper has discussed the most literatures related to GANs. But it would be better to include and discuss representative and recent literatures in hyperparameter searching and autoML, as the learning of distribution parameters can be regarded as an instance of hyper-paramter searching and the Gumbel-Softmax trick has been widely used in autoML methods (e.g., FBNet [Wu et al., 2018] and SNAS [Xie et al., 2018]).

Reproducibility: Yes

Additional Feedback: --- Review Update ----- Thanks for the authors' feedback. I hope the authors could include the discussion and clarification of the rebuttal in the main paper and improve the draft according to all the reviews.


Review 3

Summary and Contributions: The paper introduces an approach base on infoGAN for unsupervised learning of disentangled representations on class-imbalanced data. The proposed model can be though as a modification to infoGAN where the assumption of uniform distribution for the categorical latent is dropped. In order to do that the authors used the Gumbel-Softmax trick to learn the distribution of the categorical latent. In addition contrastive learning is used to enforce that categorical factors are modeled by the categorical latent. Experimental results on mnist, synthetic data of two independent object categories (cars and chairs) and real data (faces) show the usefulness of the approach.

Strengths: The main strength of this paper is the value of the studied problem. Unsupervised learning of disentangle factors for class-imbalanced datasets has a potentially great impact on the field. To my knowledge this is the first paper studying that problem. The results on simple synthetic data are legitimate.

Weaknesses: The main weakness of this paper is novelty. The paper builds off of infoGAN and uses to known components from previous work (Gumbel softmax and contrastive loss) to adapt it to the class-imbalanced scenario. While I agree that the components help mitigate the problem it seems to me that there's no design novelty. If there's a novelty in this paper is to use those known components (without any modifications) together with infoGAN and applying that model to the class-imbalanced problem. In addition the experiments seem legitimate but small datasets are used (eg. mnist). Where the class-imbalanced problem is recreating by sampling the original data. Only one realistic dataset is used in the experiments (YT-Faces) and while the quantitative results show an improvement over standard infoGAN the qualitative results show that the model tends to mode-collapse. I would be convinced of the proposed method value if the model would have been used for example to model ShapeNet categories (which are highly imbalanced) where the model learns a categorical latent at the object category level (eg. cars vs chairs vs planes ....) rather than at the object sample level (eg. sedan vs pickup).

Correctness: The claims for the method and the empirical section are correct.

Clarity: The paper is clearly written and easy to understand, I want to congratulate the authors for not making it arbitrarily complex.

Relation to Prior Work: Prior work is properly discussed and relevant citations are provided. The discussion about the failure modes of infoGAN on class-imbalanced data is elegant and easy to understand.

Reproducibility: Yes

Additional Feedback:


Review 4

Summary and Contributions: The paper proposes improvements on InfoGAN for learning disentangled representations when trained on imbalanced data. The main improvements are: (i) use of Softmax-Gumbel distribution to learn the distribution of classes (specifically identities), (ii) enforce representation invariance to image transformations through contrastive loss.

Strengths: - Unsupervised learning on imbalanced datasets is an important and under-explored problem - The proposed techniques of addressing class imbalance are novel, technically sound and well designed. - Code for MNIST experiments is provided in the supplementary.

Weaknesses: - All experiments are done on images with a relative small resolution (64x64).

Correctness: The method design is technically sound.

Clarity: The paper is generally well written, but there are a few places it could be improved: - More introduction to the InfoGAN loss. For example, some terms are never defined: V_GAN, the lower bound of I(c, G(c,z)). What does "ntxent" imply? Some kind of "ntx" entropy?

Relation to Prior Work: The relation to InfoGAN is well discussed.

Reproducibility: Yes

Additional Feedback: - On Line 150 it is said that penultimate layer activations are used for cosine similarity in Eq (3). In code, it seems to be activations after Leaky ReLU nonlinearity, which also are normalized before passing to cosine similarity. If it was ReLU nonlinearity, then it would produce bias for cosine similarity distance as the unit-length vectors f all have non-negative values at every dimension. Please elaborate that in the text for better reproducibility. --- Thank you for the rebuttal. Regarding datasets: comparability against previous work is important, however I think the authors could have experimented with more real-world datasets. Shapenet is definitely a step towards more realistic datasets. If authors are keen to try, here is a real-world dataset with significant class imbalance: http://www.vision.caltech.edu/visipedia/CUB-200.html

[Author Response · NeurIPS 2020]

| | MNIST | | YTF | | 3D-cars | | 3D-chairs | |
|---|---|---|---|---|---|---|---|---|
| | NMI ↑ | ENT ↓ | NMI ↑ | ENT ↓ | NMI ↑ | ENT ↓ | NMI ↑ | ENT ↓ |
| JointVAE ($\beta = 10$) | $0.536 \pm 0.13$ | $1.032 \pm 0.29$ | $0.372 \pm 0.06$ | $1.751 \pm 0.03$ | $\mathbf{0.452 \pm 0.24}$ | $1.026 \pm 0.43$ | $0.392 \pm 0.28$ | $2.053 \pm 0.46$ |
| JointVAE ($\beta = 20$) | $\mathbf{0.704 \pm 0.08}$ | $\mathbf{0.661 \pm 0.13}$ | $0.421 \pm 0.04$ | $1.687 \pm 0.02$ | $0.441 \pm 0.31$ | $\mathbf{1.022 \pm 0.34}$ | $0.431 \pm 0.26$ | $\mathbf{1.817 \pm 0.42}$ |
| JointVAE ($\beta = 30$) | $0.680 \pm 0.07$ | $0.701 \pm 0.13$ | $0.447 \pm 0.03$ | $1.717 \pm 0.02$ | $0.391 \pm 0.23$ | $1.082 \pm 0.53$ | $0.448 \pm 0.21$ | $1.909 \pm 0.44$ |
| JointVAE ($\beta = 40$) | $0.676 \pm 0.09$ | $0.713 \pm 0.14$ | $\mathbf{0.479 \pm 0.02}$ | $\mathbf{1.662 \pm 0.03}$ | $0.324 \pm 0.45$ | $1.151 \pm 0.51$ | $\mathbf{0.480 \pm 0.24}$ | $1.986 \pm 0.41$ |
| JointVAE ($\beta = 50$) | $0.649 \pm 0.09$ | $0.774 \pm 0.15$ | $0.435 \pm 0.03$ | $1.695 \pm 0.02$ | $0.376 \pm 0.40$ | $1.193 \pm 0.25$ | $0.377 \pm 0.31$ | $2.120 \pm 0.30$ |
| InfoGAN + $L_{ntxent}$ | $0.838 \pm 0.05$ | $0.351 \pm 0.09$ | $0.712 \pm 0.01$ | $0.831 \pm 0.02$ | $0.617 \pm 0.28$ | $0.835 \pm 0.52$ | $0.438 \pm 0.15$ | $1.237 \pm 0.32$ |
| Ours | $\mathbf{0.889 \pm 0.04}$ | $\mathbf{0.213 \pm 0.09}$ | $\mathbf{0.792 \pm 0.01}$ | $\mathbf{0.636 \pm 0.01}$ | $\mathbf{0.850 \pm 0.07}$ | $\mathbf{0.303 \pm 0.15}$ | $\mathbf{0.650 \pm 0.08}$ | $\mathbf{0.765 \pm 0.18}$ |

Thank you for the helpful comments. We are encouraged that the reviewers found the problem setting important &
unexplored (R2,3,4), and our solution effective & reasonable (R1,2,4) in overcoming issues of existing work (R1,3).

**[R1] Clarifications regarding the VAE-based baseline** We apologize for not including the hyperparameter details
of JointVAE. We use the KL term for both continuous as well as discrete variables, to follow the standard normal
($\mathcal{N}(0, 1)$) and uniform categorical distribution (Cat($p = 1/k$)), respectively. We use uniform categorical because of
the unsupervised nature of the problem (L53-5, 83-4), and our full approach itself starts from uniform initialization
(L127-30 supp). We use the same weight ($\beta$) for both KL loss terms (similar to JointVAE paper), the value of which
was first decided empirically based on image reconstruction quality - we observed that a value in 100s (e.g. 100-300)
resulted in poor reconstruction quality. We ultimately went with $\beta = 30$ (the value chosen by JointVAE for MNIST)
for all datasets. We present an ablation study on the effect of strength of the KL term ($\beta$) on disentanglement in the
table above. While we agree that a lesser weight on the KL term might imply lesser restriction for inference model to
follow the uniform prior, it might result in reduced disentanglement as well. So, starting from a particular value (say
$\beta = 30$), it is not clear whether increasing (towards uniform) or decreasing (towards less disentanglement) it would help
from disentanglement's point of view. We observe this in the ablation study as well, where low and high values of the
weight ($\beta = 10$ and $\beta = 50$ respectively) usually result in low disentanglement scores. We can, however, get slightly
better results with alternate $\beta$s (different $\beta$ for different datasets) than the ones reported in the main paper, and we'll
update them with these in the final version. Note that for our approach we don't perform an exhaustive search for $\lambda_2$ (in
$L_{final}$); it's set as 10 for all datasets (L124-5 in supp). Finally, comparison to FactorVAE is not directly applicable, as it
can only capture continuous factors, and the paper itself mentions the inability to capture discrete factors as a limitation.

**[R2] [R3] Concerns regarding technical novelty** We agree we leverage existing techniques (L166-8). However, they
have previously been used in orthogonal areas: Gumbel-softmax was introduced for differentiable sampling of one-hot
like variables, and identity preserving transformations have been used as part of data augmentation, avoiding overfitting,
representation learning, etc. In this work, we've integrated these techniques in a coherent framework to address an
important problem in a *novel* setting of learning disentangled representations in class-imbalanced data (L169-70).

**[R3] [R4] Concerns regarding datasets** We'd like to point out that seminal works in learning disentangled rep-
resentations (e.g. InfoGAN, $\beta$-VAE) present results on such small datasets, where it is relatively easy to ac-
count for, and capture the factors of variations. R3 states "..model tends to mode collapse on YTF" - we re-
spectfully disagree: the categories in YTF consist of video frames of the same person, resulting in very similar
*real images* themselves. Faithfully modeling such image distribution hence results in similar generated images.

Uniform-InfoGAN

Ours

R3 further suggested to try the ShapeNet dataset, which is naturally imbalanced. The original dataset was too big to operate during the rebuttal phase ($\sim$ 600k images, with 10 renderings per model), so we created a subset consisting

39 of 5 categories - cars, airplanes, bowl, can, rifle - in a way which maintains the original imbalance between categories.
40 Due to time constraints, we're only able to compare Uniform-InfoGAN and our final method. We can see that InfoGAN
41 (NMI: 0.545, ENT: 0.687) mixes up different categories more frequently (rows 1/3) than our method (NMI: 0.781, ENT:
42 0.432), which is more consistent when grouping same category instances together. We'll include more analysis.

43 **[R2] Clarifications regarding training components** The first component, Gumbel-Softmax, should be thought of as
44 making the latent distribution flexible, so that the model can capture *any* discrete factor having $k$ modes (not necessarily
45 object identity) present in any ratio. $L_{ntxent}$'s role is to push the discovered factor to better correspond with object
46 identity (L162-3), in balanced or imbalanced case (as R2 points out). We include the results for Uniform-InfoGAN +
47 $L_{ntxent}$ in the table above. The results improve compared to Uniform-InfoGAN, but having the rigid uniform prior still
48 results in worse performance compared to our approach. Hence, Gumbel-softmax alone *shouldn't* be thought of as an
49 improvement, as it works best when used along with $L_{ntxent}$ (L229-33). Furthermore, we cannot use class labels since
50 this is an unsupervised task. Finally, we'll discuss the mentioned related works; we thank R2 for pointing them out.

51 **[R4] Clarification regarding input for $L_{ntxent}$** We agree and will discuss this in detail. *ntxent* is described in L159.

[Meta-Review · NeurIPS 2020]

This is a well written paper, and most concerns were addressed in the well written rebuttal. The main remaining suggestions are perhaps to add another realistic dataset for more complex experiments as well as more analysis is needed on GT class distribution + InfoGAN not performing well.